# The Sustainability of Superior Performance of Platform Complementor: Evidence from the Effects of Iterative Innovation and Visibility of App in iOS Platform in China

**Na Wang, Shuangying Chen** **, Lei Xiao \*** **and Feng Fu**

School of Economics and Management, University of Electronic Science and Technology of China, Chengdu 611731, China; 201921150213@std.uestc.edu.cn (N.W.); shychen@uestc.edu.cn (S.C.); fufeng@std.uestc.edu.cn (F.F.)
\* Correspondence: xiaolei@uestc.edu.cn (L.X.)

**Abstract:** The sustainability of superior performance is a vital goal for complementor in mobile platform ecosystem, but it is becoming increasingly difficult to realize it. The previous literature about the sustainability of superior performance is insufficient and lacks deep research the mechanisms that affect the sustainability of superior performance. Based on user-oriented perspective, this study examines the direct and interactive effects of app's iterative innovation and visibility on the sustainability of complementor's superior performance. Using a two-way fixed effect model and monthly panel data, with the final dataset of 4596 observations from 384 apps of top-ranking positions by revenue available from December 2018 to December 2019 on the Apple iOS App Store in China, the results show that a) app's iterative innovation and visibility (including platform recommendations and download rankings) promote the sustainability of app's superior performance; b) platform recommendations and iterative innovation have a synergistic effect on the sustainability of app's superior performance; and c) the download ranking has a substitution effect on iterative innovation to promote the sustainability of app's superior performance. The findings of this study have theoretical and practical implications for improving the sustainability of app's superior performance in the digital age.

**Keywords:** sustainability of superior performance; iterative innovation; platform recommendation; download ranking; user-oriented

## 1. Introduction

While the sustainability of superior performance is a vital goal for complementor in mobile platform, it is becoming increasingly difficult to realize it [1]. Mobile platform provides not only the important entrepreneurial opportunities for complementor, but also the hypercompetitive environment for sustaining superior performance. For example, iOS platform is orchestrated by Apple (the platform firm) and app (complementor), which interact with the specific handset and operating system combination offered by Apple. In 2019, according to the Ministry of Industry and Information Technology in China, the total number of Chinese apps in the Apple iOS platform exceeded 1.5 million. Moreover, the number of apps submitted on the iOS platform about 30,000 and the number off the platform is over 70,000 per month.

Consequently, with more and more apps (complementors) flooding mobile platform, the sustainability of apps' superior performance is challenged by the following issues: (1) How can app meet user demands and improve user experience quickly to sustain its superior performance in mobile platform, where user needs are highly volatile and user conversion costs are low? (2) How can app obtain new users and stick to old users to sustain superior performance, under the contradiction between the scarcity of users' attention and the massive supply of app? These challenges highlight the importance of

continuously focusing on users' needs and improving user experience, as this drives the sustainability of superior performance from a user-oriented perspective. However, the literature about the sustainability of superior performance is insufficient and lacks deep research the mechanisms that affect the sustainability of superior performance from the user-oriented perspective.

In response to the first challenge, this study focuses on app's iterative innovation with users' perspectives. Iterative innovation means continuously providing users with more differentiated services and a better user experience and responding to users' highly variable and complex needs. For example, in 2019, the iterative innovation of Wechat, Taobao, Iqiyi and many other apps were more than one time every month on average. In other words, app's iterative innovation means that app must sustain to expand the app's functions or provide new content quickly that meets the unsatisfied needs of both existing users and potential users, thus providing new services and improving the user experience [1]. These new services take the new functions and content of the app as the carrier. Iterative innovation has attracted a great deal of attention from scholars [2,3]. However, these studies have failed to address how an app can respond to the dynamic needs of app users and thus achieve a value-added.

In response to the second challenge, this study pay attention to app visibility. In a mass-supply setting, visibility can be used predictably to capture scarce user attention, acquire new users, and retain old users. Here, app visibility refers to the possibility that a potential user might discover the app [4]. The literature on app visibility is based mainly on market phenomena and focuses on the contradiction between the explosive growth of apps and the scarcity of user attention [5]. It also addresses the resulting increase in user search costs and the challenge of accumulating users from scratch. The previous literature focuses on app visibility as a single dimension, and does not distinguish between platform recommendations and download rankings. These studies examine the effect of download rankings on the success of apps in the market [4,6] or the design of the platform recommendation system [7,8]. However, studies of app visibility have not treated platform recommendations and download rankings as two dimensions of an app visibility construct that could further reveal the mechanism of user attention.

Notably, when an app has low visibility, it is difficult not only to attract new users and retain old users but also to display the new services provided by iterative innovation to potential users. In other words, iterative innovation cannot be successful without the potential users' discovery of and interest in the innovation [4]. In contrast, high visibility increases the possibility that iterative innovation will be discovered by potential users. Therefore, the sustainability of app's superior performance requires each iterative innovation to meet the needs of potential users and may also be affected by the app's visibility. Therefore, the complex mechanism by which both app's visibility and iterative innovation affect the sustainability of app's superior performance is worthy of further study.

Thus, the aim of this study is to explore the direct impact and interactive impact of iterative innovation and visibility on the sustainability of complementor's superior performance. Adopting a user-oriented perspective, we assembled a monthly panel dataset of top-performing app in the iOS over 13-months period. Following the prior literature [1], the analysis is based on the condition that app sustained superior performance by continuing to be in the top-performance stratum in mobile platform ecosystem (i.e., Top 500 apps by revenue). Based on the panel dataset, this study explores how app's iterative innovation and visibility—including platform recommendations and download rankings—directly affect the sustainability of app's superior performance. In addition, we also explore the interaction on the sustainability.

The remainder of the paper is organized as follows: Section 2 is about the theory of user-oriented perspective. Section 3 presents the research hypothesis and provides deductive details. Section 4 describes the data, sample, and variable construction. Section 5 presents the main empirical results of regression using two-way fixed effects models

and conducts robustness tests. Section 6 summarizes the research conclusions, research contributions and future research directions.

## 2. Theoretical Background

User-oriented perspective, which originated in the field of business management, has transformed marketing from a product-oriented practice to a user-oriented practice. User orientation is essentially a series of values that prioritize the interests of users [9]. These values emphasize the need to always attend to users' needs, because an enterprise's competitive advantage and performance growth derive from the creation and maintenance of user value [10,11]. With the evolution and development of user-oriented theory, the focus of scholars has shifted gradually from commodity-oriented to service-oriented logic. From this latter perspective, value is no longer created by the enterprise alone, but by the enterprise and user together [12].

A user-oriented perspective emphasizes that users are the leaders in value creation. Accordingly, to increase the value capture space and improve performance, enterprises must provide conditions for user value creation [10,13]. A user purchases a service to obtain value from the utility or convenience provided by the service, and therefore the user's experience and emotions determine their purchase intention [14,15]. In the context of the mobile Internet, innovation and value creation are dominated by the service and user experience provided by apps (as a typical experiential product). Here, users are the key drivers of network effects, traffic flow and economic benefits [16,17]. Therefore, meeting users' needs and improving their experience are the foundation from which app performance can be enhanced and superior user value can be created and maintained.

Relevant to user perspective, demand-side strategy also emphasizes value creation for users as the necessary condition for value capture [18]. Value creation focuses on increasing the benefits of the user group [19,20]. Under the demand-side strategy, value creation is giving consideration to users and their dynamic, heterogeneous, endogenous, and, at times, potential demands of users. Thus, demand-side strategy implies that value creation for users is a prerequisite for firm success [18].

User value is created by increasing user benefits or reducing user costs [21]. Innovation research taking a demand-side approach often emphasizes market change and heterogeneity of user demand [22]. Based on the user-oriented perspective, user benefits in the app business can be increased mainly by continually developing new content or new functions that address users' heterogeneous needs, i.e., by responding to users' needs in a timely manner and continuously improving the app through iterative innovation. This coincides with the findings of Kalcheva et al., that is, a shift in demand has an impact on innovation [23]. In addition, improving the market visibility of the app can effectively reduce the search cost, thereby creating user value. Therefore, improving the iterative innovation and the visibility of an app are important paths toward user value creation and the sustainability of app's superior performance in the app market.

From the user-oriented perspective, the goal of iterative innovation is to improve the user experience while continuing to meet users' needs, i.e., to create user value continuously. In the field of innovation research, iterative innovation, which means 'rapid change', has become an effective innovation model in a highly uncertain environment [24]. Successful innovation requires the resolution of the contradiction between change and users' needs [25], while failed innovation usually stems from the ignorance of user needs. For example, Yik Yak, once valued at 400 million US dollars. However, because its anonymous mode threatens network security, the platform reduces its visibility. Subsequently, Yik Yak abandoned the anonymous mode in the new version in order to attract new users, which resulting in the loss of a large number of old users. Yik Yak's iteration accelerated its failure. The reason is that Yik Yak has no clear understanding of users' heterogeneity requirements. Priem points out that inability to identify users' future needs will hasten failure when faced with disruptive innovation [22]. Not all iterative innovation can be successful. Developers have different resources, capabilities and strategies, but the focus of user needs is the key to

success [26]. It must be clear that there are two different outcomes of iterative innovation: success and failure, and the failure of iterative innovation decisions will make the app unable to generate profits [27].

Consequently, developers must prioritize user value and continue to create and meet users' needs through iterative innovation, which is based on identifying, analyzing and understanding these needs. We can only generate excess profits by satisfying users' diverse and dynamic value demands; this requires continuous iterative innovation process that provides users with new functions and services [11].

However, before a provider can improve the user experience and meet the needs of users, it must first have users. The app must be visible before it can be used. App visibility reflects an app's exposure in the application market. In the highly competitive digital market environment, apps need to increase information interaction with users. This is because the user must see and believe an app's information before using it. The 'visibility' paradigm means that visibility to the public increases through information [28]. Improving app visibility can increase users' exposure to the app and enable them to receive more information, thereby generating attention and increasing app use [6,29]. Improving app visibility can make it more convenient for users to obtain apps, reduce potential user search costs and improve user satisfaction [6]. Furthermore, improving app visibility attracts new users, which triggers network effects and increases the perceived value and experience among existing app users [30,31]. Therefore, from the user-oriented perspective, improving app visibility will improve the user's experience and value.

### 3. Research Hypothesis

*3.1. Iterative Innovation and the Sustainability of App's Superior Performance*

Iterative innovation provides app's users with additional incremental services that realize immediate adjustments and respond to changes in users' needs, meet unmet needs or even create needs [32]. From the user-oriented perspective, iterative innovation can add new user value and improve the app experience, thereby attracting new users, retaining existing users and promoting consumption by new and old users. These functions are beneficial to the sustainability of app's superior performance for the following reasons.

Iterative innovation can attract the attention of new users and enhance their perception of the app's value, which is translated into purchases or consumption. First, iterative innovation can provide users with new services, increase the attractiveness of the app, stimulate interest and attention and promote use [1]. Second, iterative innovation will significantly differentiate the services provided by the app from those provided by competitors, which will enhance users' perceived value of these differentiated services and increase potential users' purchase intentions [15,33,34]. Third, by providing new services, iterative innovation can meet new users' needs, increase user satisfaction, encourage new users' purchase behavior, achieve performance improvements and add user value [32].

Additionally, iterative innovation process can increase the retention of existing users while enhancing their continuous use and purchase intentions. Iterative innovation process that emphasizes the innovation can offer new and differentiated services continuously to existing users. Accordingly, existing users will consider the app features and functions to be increasingly satisfactory, the content to be increasingly abundant and the user experience to be increasingly improved. Furthermore, these new services can also address the unmet needs of existing users and improve their satisfaction, leading to increases in user loyalty and continuous use and consumption intentions [35,36]. Therefore, the following hypothesis is proposed.

**Hypothesis 1 (H1).** *App's iterative innovation will positively affect the sustainability of app's superior performance.*

### 3.2. Visibility and the Sustainability of App's Superior Performance

App visibility reflects the exposure of an app in the market and has become a decisive factor in success [4]. Platform recommendations and user download rankings are indispensable dimensions of app visibility. These dimensions differ substantially. A platform recommendation refers to a subjective but authoritative endorsement by the platform editing team, whereas a download ranking is an objective measure of user download behavior. Accordingly, the former can more effectively solve the "cold start" problem faced by a new app, while the latter has a strengthening effect on well-known apps with an established user base. Therefore, we use both platform recommendations and download rankings to characterize app visibility and study the effects of these two dimensions on the sustainability of app's superior performance.

#### 3.2.1. Platform Recommendation and the Sustainability of App's Superior Performance

Users may find it difficult to identify apps of interest in mass mobile application settings. To reduce users' search costs, the Apple iOS App Store created a user orientation-based application recommendation mechanism [37]. Platform recommendation is a key channel through which a less well-known app with a weak user base or a new app on the market can become more visible and attract potential users. Furthermore, an authoritative platform recommendation can increase the retention and loyalty of existing users. This study is based on the belief that platform recommendations not only reduce users' app search and acquisition costs but also enhance the authority of app information and user growth. Thus, platform recommendation is vital to the sustainability of app's superior performance for the following reasons.

Platform recommendation encourages potential users to use or consume apps by reducing the app search and acquisition costs. This dimension increases the app's exposure, thereby attracting attention and increasing the likelihood of discovery by potential users [6]. Simultaneously, platform recommendation reduces the app search cost [38] and thus enables potential users to quickly and effectively obtain the needed product. This will encourage potential users to use the app to obtain satisfaction.

Due to their authoritative nature, platform recommendations also increase the retention of existing users and encourage them to continue to purchase the app's services. The low cost of user conversion encourages existing app users to try new apps constantly. Therefore, existing apps are challenged by low user retention. Platform recommendations are an authoritative and objective form of third-party certification. Users perceive apps that receive more platform recommendations to be more useful [39], reliable and trustworthy [40]. This trust enhances the perceived value of the app, enhances the 'lock-in' effect on existing users, increases user retention and promotes continued purchases of app services. In summary, we propose the following hypothesis.

**Hypothesis 2 (H2).** *Platform recommendations can promote the sustainability of app's superior performance.*

#### 3.2.2. Download Ranking and the Sustainability of App's Superior Performance

A download ranking reflects the user's objective download behavior and is another dimension used by application platform operators to increase app visibility. Although a download ranking is published by the platform, it is determined objectively by user downloads. The higher the number of downloads, the higher the popularity of the app among users and the higher the download ranking. Because users tend to browse downward from the top of the app list, the top-ranked apps are more visible and attract more attention [4]. The effect of download rankings on the sustainability of app's superior performance is exerted mainly by its influence on the consumption decisions or behaviors of potential and existing users. The specific reasons are as follows.

Apps with the highest download rankings attract more attention from potential users. These apps also increase potential users' expected utility, thereby prompting them to download and experience the (free or paid) app's services. According to the first-cause effect of selection [41], most users start browsing from the top of the app list. Additionally, the 'same-screen competition' feature prevents users from downloading too many similar apps. Furthermore, apps have network externalities, the existence of which prompts users to choose mainstream products in the market [30,33]. Network externalities are defined as the increasing utility that a user derives from consumption of a product as the number of other users who consume the same product increases [42]. Therefore, the apps with the highest download rankings are expected to provide superior expected utility and experiences, and these features encourage potential users to download, use and purchase the apps.

Apps with top download rankings can also further enhance the network effect, thus increasing the utility of the app for existing users. First, user utility depends on each other. With the increase of the number of users, the network effect of an app is enhanced, and the user value increases. The ability of apps with top download rankings to attract and acquire new users has led to the emergence of a self-reinforcing positive feedback mechanism and an exponential increase in the number of users for top downloaded apps, which has enhanced the network effects [43] and thus user value and utility. Second, existing app users who enjoy increasing value and utility as a consequence of the ever-increasing network effect will remain loyal to the app [30,31,33]. Together, these arguments inform the following hypothesis.

**Hypothesis 3 (H3).** *The download ranking can promote the sustainability of app's superior performance.*

### 3.3. Interaction between App's Iterative Innovation and Visibility

The above hypotheses suggest that app's iterative innovation and visibility can each promote the sustainability of app's superior performance. From the perspective of user orientation, iterative innovation improves the user experience and continuously creates and meets users' needs, whereas visibility is a necessary condition of app use. It is possible that these characteristics interact with each other.

It is worth mentioning that platform recommendation and download ranking can also increase the visibility of apps, but their mechanisms are likely to be different. Because most of the apps recommended by the platform are approved by the editorial team, so the recommended apps are usually new or significantly updated [6]. However, download ranking is more related to the objective download behavior of users [44]. Consequently, we believe that users are more confident in well-known apps with high rankings [45]. High rankings are the result of users' objective download behavior, indicating that these apps have gained the preference of many users and will have more powerful functions to meet user needs [46]. Therefore, users will have higher expectations for apps with higher download ranking [47,48]. However, the apps recommended by the platform on some topics are usually new and little-known [6]. Users tend to think that the functions and experiences that can be obtained from these apps are uncertain. Thus, users generally have a sense of freshness in them, but they do not have high expectations.

3.3.1. Synergistic Effect of Platform Recommendation and App's Iterative Innovation

Here, we argue that a platform recommendation and app's iterative innovation have a synergistic positive effect on the sustainability of app's superior performance for the following reasons.

For an app, platform recommendation enhances the effect of iterative innovation on the purchase behavior of potential users. In this regard, a high level of iterative innovation can attract more potential users and encourage their use and purchase of the app services.

However, iterative innovation provided by the new iteration of the app must be observed by potential users, as this is required to encourage their curiosity, interest, attention and use. The usefulness of an innovation determines the purchase behavior and satisfaction of the user and the perceived usefulness and value of the product [49]. Therefore, iterative innovation promotes the use and purchase behavior of the potential app users, but this relationship must depend on the visibility of the innovation. A platform recommendation is an official and authoritative form of certification and a marketing mechanism; in this context, a recommendation increases the possibility that an app will be discovered by potential users [6]. Therefore, an increased number of platform recommendations increases exposure and allows more potential users to discover the new services added to the app by iterative innovation. Accordingly, the recommendation increases the attractiveness of the app to potential users, encourages the use of the app to satisfy their needs and promotes their purchase behavior.

Additionally, the authority of a platform recommendation enhances the effect of the iterative innovation on the purchase behavior of existing users. An increased number of authoritative platform recommendations enhances the existing users' perceptions of the value and reliability of the app [40]. In turn, the existing users are more willing to use the new services provided by iterative innovation, which will satisfy their demands and demonstrate the superiority of the new services.

In general, platform recommendations arouse the curiosity of new users and increase the perceived value of existing users, thereby promoting users to use the services provided in iterative innovation to meet their needs. Given these arguments, the following hypothesis is proposed.

**Hypothesis 4 (H4).** *Platform recommendation and app's iterative innovation have a synergistic positive effect on the sustainability of app's superior performance.*

### 3.3.2. Substitution Effect of App's Download Ranking on Iterative Innovation

We also argue that app's download ranking has a substitution effect on iterative innovation to promote the sustainability of app's superior performance for the following reasons. For an app, a higher download ranking weakens the positive effects of app's iterative innovation on new user satisfaction and the sustainability of app's superior performance. Download ranking is an objective measure that helps users to select app. New users expect a better experience from top-ranked app and thus have higher requirements and expectations regarding the app functions, content and experience. According to the satisfaction formation process, as explained by the expectation theory [50], user satisfaction is reduced when the expectation is greater than the perceived value. Therefore, the high expectations of new users for app with the highest download rankings will reduce the effect of iterative innovation on new user satisfaction [51,52]. Consequently, iterative innovation to promote the sustainability of app's superior performance will also be reduced.

Moreover, a higher download ranking may weaken developers' engagement in the iterative innovation process. This would hinder their responsiveness to existing users' needs and thus reduce the ability of iterative innovation to promote the sustainability of app's superior performance. As download ranking objectively reflects the app's popularity among users, a higher download ranking can effectively promote further increases in the number of users. In addition, apps with the highest download rankings tend to yield better performances. Therefore, apps with higher rankings are subject to lower levels of survival or development pressure, which may reduce developers' efforts toward further iterative innovation [53]. Consequently, iterative innovation would decrease, resulting in an insufficient response to the existing users' needs, user churn and a reduction in the sustainability of app's superior performance.

In general, higher download rankings tend to increase the expectations and requirements of new users and reduce the development efforts of developers, thereby weakening

the effect of iterative innovation on app's sustainable performance through these two aspects. Therefore, the following hypothesis is proposed.

**Hypothesis 5 (H5).** *App's download ranking has a substitution effect on iterative innovation to promote the sustainability of app's superior performance.*

In summary, we propose five hypotheses, and the research model of the study is shown in Figure 1.

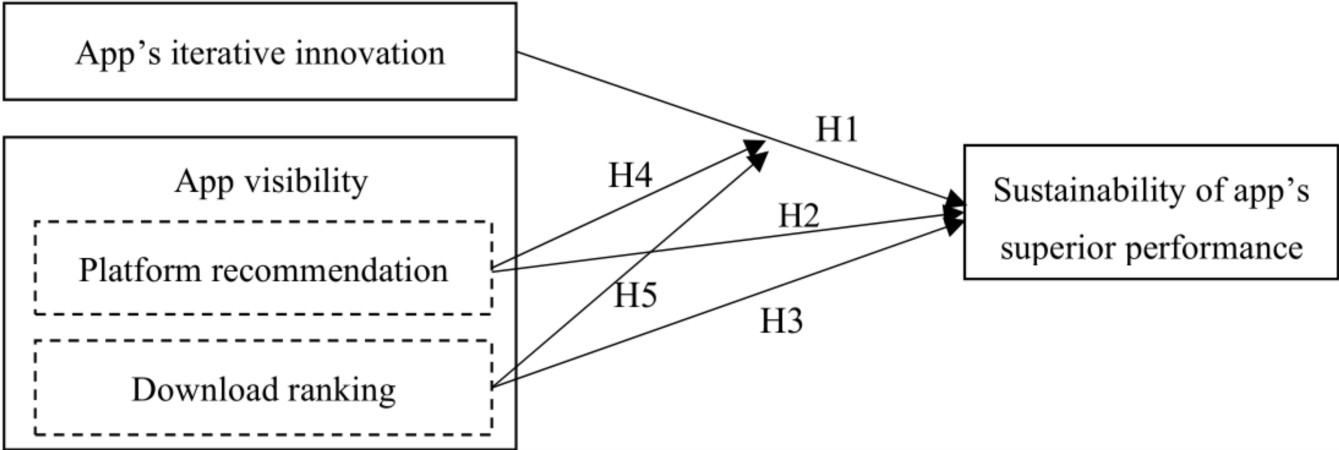

**Figure 1.** Research model.

## 4. Methodology

### 4.1. Data

The primary sources for our data are Qimai Data (www.qimai.cn, accessed on 22 May 2020), a domestic professional mobile marketing promotion data analysis platform in China. It provides users with multi-dimensional and objective historical data from the iOS and Android app stores and real-time queries of dual-platform lists. Currently, Qimai Data includes more than 7 million apps and more than 12 million app data points. Its data are extensively used by app developers, financial analysts, and venture capital firms.

The sample for this study consisted of apps that attained top-ranking positions by revenue (i.e., Top 500 of bestseller apps) in the application category apps from the bestseller list in the Chinese version of the Apple iOS App Store from December 2018 to December 2019. We excluded apps that were removed from the Top 500 list during the research period, So the final sample covers 22 app categories. This research sample was selected for the following reasons. (1) Followed the prior research [1], a top-ranking among the top 500 apps demonstrates app's superior performance, and also ensured a stable sample size for this study. (2) The Apple iOS App Store provides richer and more concentrated data that are more suitable for the research focus of this paper, whereas the domestic Android app store has many channels and scattered data. Additionally, as of late 2019, the number of apps in the Apple iOS App Store in China accounted for approximately 41% of all apps in the domestic market. In addition, the dataset does not include game apps, due to the difference between the operations and revenues of game apps and application category apps. Specifically, application category apps are generally functional, and their functions need to be told to users in a simple and clear way. The game app focuses on entertainment and needs to provide users with certain elements of exploration. And the main revenue source of application category app is advertising, which doesn't charge users directly. However, the direct consumer group of game apps is their players. This study tends to explore the relationship between iterative innovation, visibility and performance of functional app. Therefore, game apps are not included in the sample.

We assembled a unique monthly panel data set of top-performing apps in the iOS ecosystems over the 13 months period. And our sample contained the monthly data of 384 apps in 22 categories, including education, business, entertainment etc., thus providing a balanced panel of 4992 observations. We performed truncation and replaced the top 1% and bottom 1% of values with critical values and obtained a final set of 4596 valid samples. It should be noted that this study attempts to explore the sustainable performance of application category apps. Therefore, although we get the data of 22 different app categories from the list of app stores, we regard them as application category apps, and do not explore the differences in sustainable performance of different app categories.

To ensure the authenticity and reliability of our data from Qimai, we also collected the data on app's iterative innovations on Kuchuan (https://www.kuchuan.com/, accessed on 4 June 2020) as a supplementary data source (Supplementary Materials) and checked these with the data collected from Qimai for verification. Note that both Qimai and Kuchuan do not generate their own data but accumulate daily data from Apple App stores over time and offer their users easy-to-use tools for analyzing trends.

### 4.2. Measurement

The section is about the measurement of all variables in this study. The names of the variables and their measurements are showed in Table 1.

**Table 1.** Descriptions of variables included in the analysis.

| Variables | Measure |
|---|---|
| Dependent variable | |
| Sustainability of app's superior performance | The daily weighted average of the bestsellers list ranking in the total chart for the month |
| Independent variables | |
| Platform recommendation | The number of days the app was recommended by the platform in a given month |
| Download ranking | The average download ranking of apps from the total chart for a given month |
| Iterative innovation | The number of new functions and new content in a monthly app update |
| Control variables | |
| Developer operational capacity | The number of apps that developers still operate in a given month |
| Developer experience | The interval (in months) from the developer's first app release to the current month |
| Number of reviews | The number of user reviews in a given month |
| Platform competition | The number of apps that iOS is still operating to the current month based on the category of app |
| Age (of the app) | The interval (in months) from the app release to a given month |
| Popularity | The number of days that the app is ranked in the top 300 on the bestseller list of application chart for a given month. |
| Sustainable potential | The number of days that the app is ranked in the top 500 on the bestseller list of total chart for a given month. |

Notes: 1. Apple App store has three types of charts: total chart, application chart and game chart. The total chart includes all app categories for ranking; the application chart only includes application category apps for ranking; the game chart only includes game apps for ranking. (Except for game apps, other apps are application category apps); 2. There are three lists in each type of charts-free list, pay list and bestseller list. The free list and paid list are ranked based on the downloads of free and paid apps, respectively, so we refer to them as the download list for the purpose of this study. The bestseller list is ranked according to the sales revenue of the app, which including free app and paid app.; 3. Since all apps are included in the total chart, users browse the total chart more. At the same time, the total chart shows a broader competitive market for the apps in our sample. So, we use the ranking of apps in the total chart to measure sustainable performance and download ranking. However, it should be noted that we only use the ranking of the apps in our sample in the total chart. Our sample only includes application category apps, not game apps.; 4. In order to control the potential impact of the app's ranking in the application chart on its sustainable performance, we added the "Popularity" as a control variable.

### 4.2.1. Dependent Variable

The sustainability of app's superior performance. Following Zhou et al. [54], the study examines the sustainability of superior performance for app by observing that app continues to be among the Top 500 of bestseller apps by revenue in the iOS platform, and using the daily weighted average of the bestsellers list ranking in the total chart for the

month as the measure of the sustainability of app's superior performance. The app has a ranking every day, we assign scores of 5, 4, 3, 2, 1 to rankings 1–300, 301–600, 601–900, 901–1200 respectively. Then we get the performance index according to the score of the app in the current month. To reduce possible endogeneity problems, we used the sustainability of app's superior performance during a 1-month lag period as the dependent variable.

### 4.2.2. Independent Variables

Iterative innovation. Consistent with prior research [1], we used the number of new functions and new content in the monthly app update as the measure of app's iterative innovation. Because developers may update the same app several times in a month, new functions and new content are the most important features of an app digital service innovation [32]. We studied the new app features revealed in the update log. Through text mining of app update log to judge the new functions and new content in the iteration. For example, Baidu Netdisk wrote in its update log that "Notes function is online, supporting graphic editing and link collection". We regard this as a new function in iteration innovation. Since we pay more attention to the novelty dimension of app iteration, we think that the "improving system stability" mentioned in the update log is an optimization based on the original, not as the number of innovations. In addition, we think it is an iterative innovation to delete useless functions that are not preferred by users, although the app's update log in this sample doesn't mention the deletion of certain functions.

Platform recommendation. Following Liang et al. [6], this study used the number of days per month during which the app was recommended by the platform to measure this variable. The higher the number of days of platform recommendation, the more users will notice and become interested in the app.

Download ranking. Following Zhou et al. [54], this study used the average download ranking of apps in the indicated month's total chart to measure this variable.

### 4.2.3. Control Variables

To control for other factors and improve the validity of the research conclusions [1,4,46], we selected the following control variables at the app, user and platform levels. At the app level, this study separately controlled for developer operational capacity, developer experience, age of the app, category, sustainable potential and popularity. At the user level, we controlled for the number of online reviews left by users. At the platform level, we controlled for platform competition.

## 5. Results

### 5.1. Descriptive Statistics and Correlation Analysis

Table 2 reports the descriptive statistics of all of the variables. And Table 3 shows the correlation analysis of the variables. All of the explanatory variables were significantly correlated with the explained variable (the sustainability of app's superior performance), indicating the rational selection of variables. In addition, the correlation coefficients between all of the explanatory variables did not exceed the critical value of 0.5, and the variance expansion factor (VIF) of the model was less than 3, indicating that there was no collinearity in this study.

**Table 2.** Descriptive statistics of variables.

| Variables | Mean | SD | Min | Max |
|---|---|---|---|---|
| 1. Sustainability of app's superior performance | 26.170 | 16.929 | 0 | 50 |
| 2. Platform recommendation | 4.047 | 9.636 | 0 | 31 |
| 3. Download ranking | 523.244 | 609.514 | 1 | 1587.677 |
| 4. Iterative innovation | 1.648 | 2.609 | 0 | 13 |
| 5. Developer operational capacity | 11.916 | 21.642 | 1 | 104 |
| 6. Developer experience | 72.491 | 29.713 | 10 | 129 |
| 7. Number of reviews | 0.072 | 0.252 | 0 | 2.040 |
| 8. Platform competition | 4.780 | 5.111 | 0 | 29 |
| 9. Age | 56.210 | 29.038 | 6 | 118 |
| 10. Popularity | 16.101 | 14.285 | 0 | 31 |
| 11. Sustainable potential | 9.138 | 13.360 | 0 | 31 |

Notes: 1. Some variables have no units which are absolute values: Sustainability of app's superior performance; Download ranking; 2. The units of the other variables are as follows: Platform recommendation (unit: day); Iterative innovation (unit: pieces); Developer operational capacity (unit: pieces); Developer experience (unit: month); Number of reviews (unit: ten thousand); Platform competition (unit: day); Age (unit: month); Popularity (unit: day); Sustainable potential (unit: day).

**Table 3.** Correlation analysis of variables.

| Variables | 1 | 2 | 3 | 4 | 5 | 6 | 7 | 8 | 9 | 10 |
|---|---|---|---|---|---|---|---|---|---|---|
| 1. Sustainability of app's superior performance | | | | | | | | | | |
| 2. Platform recommendation | 0.100 *** | | | | | | | | | |
| 3. Download ranking | 0.436 *** | 0.146 *** | | | | | | | | |
| 4. Iterative innovation | 0.188 *** | 0.081 *** | 0.176 *** | | | | | | | |
| 5. Developer operational capacity | 0.154 ** | −0.002 | 0.197 *** | 0.027 * | | | | | | |
| 6. Developer experience | 0.155 *** | 0.130 *** | 0.235 *** | 0.019 | 0.404 *** | | | | | |
| 7. Number of reviews | 0.246 *** | −0.002 | 0.275 *** | 0.063 *** | 0.048 *** | 0.021 | | | | |
| 8. Platform competition | −0.057 ** | −0.038 *** | 0.179 *** | 0.023 * | 0.139 *** | 0.032 ** | 0.041 *** | | | |
| 9. Age (of the app) | 0.178 ** | 0.044 *** | 0.234 *** | 0.027 * | 0.056 *** | 0.671 *** | 0.084 *** | −0.002 | | |
| 10. Popularity | 0.887 *** | 0.060 *** | 0.355 *** | 0.172 *** | 0.114 *** | 0.108 *** | 0.196 *** | −0.064 *** | 0.125 *** | |
| 11. Sustainable potential | 0.788 *** | 0.104 *** | 0.438 *** | 0.176 *** | 0.169 *** | 0.144 *** | 0.261 *** | −0.091 *** | 0.184 *** | 0.382 *** |

Note: * $p < 0.05$; ** $p < 0.01$; *** $p < 0.001$. N = 4596.

### 5.2. Regression Results

First, the Chow test is performed to determine which panel data method or mixed least squares (OLS) method is more appropriate. Results showed that the fixed effects model is superior to the mixed effects model, and that the two-way fixed effects model was better. Second, we needed to choose between fixed effect and random effect models. The Hausman test was applied to the model, and the results identified the fixed effect model as superior to the random effect model. For the regression, therefore, we used the two-way fixed effect model to simultaneously control the influences of individual fixed effects and time fixed effects. In addition, we use the clustering robust standard errors of app individuals. One is that this method is not sensitive to heteroscedasticity and autocorrelation in the model, which improves the estimation accuracy. On the other hand, due to the correlation of the random disturbance items of the same individual in different months, the clustering robust standard error can better capture the characteristics of intra group correlation, so as to obtain the consistent estimation of the true standard error.

Table 4 shows the results of a hierarchical regression analysis to test the effects of app's iterative innovation, visibility and their interaction on the sustainability of app's superior performance.

**Table 4.** Regression results of the two-way fixed effects model.

| Variables | Sustainability of App's Superior Performance | | |
|---|---|---|---|
| Name | Model 1 | Model 2 | Model 3 |
| Iterative innovation | | 0.057 * (0.033) | 0.070 * (0.037) |
| Platform recommendation | | 0.046 * (0.025) | 0.048 * (0.025) |
| Download ranking | | 0.003 *** (0.001) | 0.003 *** (0.001) |
| Platform recommendation × Iterative innovation | | | 0.136 * (0.075) |
| Download ranking × Iterative innovation | | | −0.177 ** (0.087) |
| Developer operational capacity | 0.063 (0.110) | 0.078 (0.116) | 0.077 (0.116) |
| Developer experience | −0.507 * (0.262) | −0.509 * (0.262) | −0.511 ** (0.258) |
| Number of reviews | −0.245 (0.421) | −0.312 (0.400) | −0.308 (0.401) |
| Platform competition | −0.045 (0.109) | −0.055 (0.110) | −0.055 (0.111) |
| Age | −0.084 (0.281) | −0.114 (0.268) | −0.085 (0.266) |
| Sustainable potential | 0.354 *** (0.027) | 0.338 *** (0.027) | 0.338 *** (0.027) |
| Popularity | 0.243 *** (0.030) | 0.217 *** (0.028) | 0.217 *** (0.027) |
| Category | controlled | controlled | controlled |
| Constant | 58.779** (23.398) | 59.222 ** (22.486) | 57.758 ** (22.568) |
| Within $R^2$ | 0.257 | 0.266 | 0.268 |
| Number of observations | 4596 | 4596 | 4596 |

Notes: 1. * $p < 0.05$; ** $p < 0.01$; *** $p < 0.001$; 2. The regression in Table 4 adopts the two-way fixed effect model to control the influence of individual fixed effect and time fixed effect at the same time.

Model 1 is the baseline model that only includes control variables. Independent variables are added in Model 2, and interaction terms of independent variables are added in Model 3. Regarding the effect of app's iterative innovation, the regression coefficients of Model 2 ($\beta = 0.057$, $p < 0.05$) and Model 3 ($\beta = 0.070$, $p < 0.05$) suggested that this variable can significantly promote the sustainability of app's superior performance, and thus Hypothesis 1 was supported. Regarding the effects of the two dimensions of app visibility, the regression coefficients of Model 2 ($\beta = 0.046$, $p < 0.05$) and Model 3 ($\beta = 0.048$, $p < 0.05$) demonstrated that platform recommendations had a significant effect on the sustainability of app's superior performance. Furthermore, regarding the effect of download rankings, the regression coefficients of Model 2 ($\beta = 0.003$, $p < 0.001$) and Model 3 ($\beta = 0.003$, $p < 0.001$) showed that this variable also had a significant effect on the sustainability of app's superior performance. Therefore, Hypothesis 2 and Hypothesis 3 were supported.

Based on Model 2, Model 3 included the interactions between app's iterative innovation and the dimensions of app visibility to test how they affect the sustainability of app's superior performance. When calculating the interaction terms, we first standardized the independent variables and then multiplied them to effectively avoid multicollinearity among the interaction terms and other variables. The results of Model 3 showed that the interaction between platform recommendations and app's iterative innovation had a significant positive effect on the sustainability of app's superior performance ($\beta = 0.136$, $p < 0.05$), showing that these independent variables had a synergistic promoting effect on the dependent variable. Therefore, Hypothesis 4 was supported. Furthermore, the interaction between download rankings and the iterative innovation had a significant negative effect on the sustainability of app's superior performance ($\beta = -0.177$, $p < 0.001$), indicating that the download ranking cancelled the promoting effect of iterative innovation in this context. Therefore, Hypothesis 5 was supported.

*5.3. Robustness Checks*

Endogeneity may lead to inaccurate conclusions in this analysis. In this study, all explanatory variables lag behind the explained variables by one period. The index of the dependent variable, the sustainability of app's superior performance, was measured at a t+1 period. So, the sustainability of app's superior performance in the t + 1 period would

not affect iterative innovation and visibility of the current app in the t period. Therefore, we effectively reduce the issue of endogeneity caused by mutual causality.

To test the robustness of the above results, we used the following methods. First, we replaced the independent variable. Using the download rankings from the application chart, with the specific treatment of the replaced independent variable the same as above, we find that the regression results are basically consistent, all five hypotheses of the study were supported. Second, we conducted a subsample regression using only the data of the top 300 apps and obtained results that were not notably different than those obtained in the previous analyses. Third, we adjusted the range of truncation. Although all of the continuous variables had been subjected to upper and lower 1% tail reduction, we applied a bilateral 5% tail-truncation range to all of the continuous variables to reduce the influence of more extreme values and verify the robustness of the previous conclusion. This repeated analysis yielded results consistent with those of the previous regression. In summary, the directionality and significance of the effects of the regression coefficients of the independent variables and their interaction items on the sustainability of app's superior performance were basically consistent with the initial results. The conclusions of our analyses are robust.

## 6. Conclusions and Discussions

### 6.1. Main Findings

In this study, conducted from the user-oriented perspective, we used a two-way fixed effect model and app data from the Apple iOS App Store to study the potential effects of iterative innovation and visibility on the sustainability of app's superior performance. We obtained the following empirical results. First, iterative innovation can promote the sustainability of app's superior performance by triggering user interest, improving perceived usefulness and responding to users' needs. Second, both tested dimensions of app visibility, namely platform recommendations and download rankings, promote the sustainability of app's superior performance. Finally, the interactions of these two dimensions of app visibility with app's iterative innovation affect have different effects on the sustainability of app's superior performance. Specifically, platform recommendations and iterative innovation synergistically promote the sustainability of app's superior performance. In contrast, download rankings appear to cancel the effect of iterative innovation on the sustainability of app's superior performance.

### 6.2. Theoretical Contributions

This study makes the following theoretical contributions. First, by showing how app, as the platform complementor, can shape its sustainability of superior performance, we depart from the existing treatments of app's direct effect [3,5]. Choi studies the direct impact of the freemium strategy on the increased sales of the paid mobile apps, and obtains the positive effects [4]. Kapoor and Agarwal explored app's sustainable performance from the perspective of platform ecosystem, and found that higher ecosystem complexity and ecosystem experience can help app-developers maintain their excellent performance [46]. However, scholars only pay attention to the direct effect of factors influencing the app sustainable performance, have not studied the interaction of multiple factors. So, we introduce the interaction of independent variables into our research model. In so doing, we offer a new lens on the interactive effects between app's iterative innovation and visibility through which apps can sustain superior performance in hypercompetitive environment of mobile platform. Thus, based on user-oriented perspective, this study examines how the interactions of app's iterative innovation with the two dimensions of visibility (platform recommendation, download ranking) affect the sustainability of superior performance of platform complementor. We thus clarify the mechanisms through which these variables interact to affect the sustainability of complementor' superior performance.

Second, this study enriches theoretical research in the field of iterative innovation [1]. The prior literature mainly focuses on the speed of iterative innovation [1], and particularly on the update frequency. For example, from the point of view of iteration speed, Boudreau

found that newcomers making similar software crowded out innovation incentives [55]. In contrast, studies have not sufficiently addressed iterative innovation, which captures the changes in products or service resulting from continuous iterations. These changes are a key factor in creating user demand and value appreciation. We propose that novel iterations can continuously provide users with new incremental services and thus respond to large changes in users' needs. Our findings confirm that iterative innovation can promote the sustainability of superior performance.

Third, this study demonstrates the limitation of considering app visibility as a concept with a single dimension [4–6]. Choi et al. believed that higher ranking of app list led to higher visibility [4], while Zhu et al. argued that platforms recommendation increased visibility by enhancing app's exposure to users [5]. we propose that platform recommendation mechanisms and users' download rankings are two important and substantially different dimensions of app visibility. Thus, our work verifies the different mechanisms through which these two dimensions promote the sustainability of app's superior performance.

### 6.3. Management Implications

Our results have important practical significance and application value. First, app developers should actively undertake user-oriented app iterations, pay attention to the iterative innovation, provide users with additional incremental services through new functions and content, and identify and respond to users' needs in a timely manner.

Second, developers must aim to improve app visibility in the market, thereby gaining users' attention and interest and increasing their purchase intention. Especially, developers who launch new apps can increase exposure through platform recommendation channels to quickly accumulate users and trigger network effects. Finally, developers with high app download rankings must still actively participate in the iterative innovation process to make full use of the existing user base and performance advantages, as users always have higher expectations of top-ranked apps.

### 6.4. Limitations and Future Research

Although we have drawn some meaningful conclusions through empirical research, our study has some limitations. First, our sample only contained app data from the Apple iOS App Store. Future research may aim to collect the more scattered Android app data through other channels and thus further analyze the mechanism and contextual value of iterative innovation and visibility with respect to the sustainability of app's superior performance in both major application systems. Second, we studied the direct effect of the iterative innovation and visibility on the sustainability of app's superior performance from the user-oriented perspective, without considering whether the platform and user behavior have an intermediary or moderating effect. Future research may address the situational mechanism of platforms and user roles from the perspective of the platform ecosystem. Third, the Apple iOS App Store did not publicly disclose specific app revenue data. In this study, the sustainability of app's superior performance is therefore based on the relative app revenue ranking. Future research may capture app revenue data through digital technology or include more indicators to comprehensively measure the sustainability of app's superior performance.

**Supplementary Materials:** The data of app's iterative innovation this paper is available online at https://www.kuchuan.com/.

**Author Contributions:** Conceptualization, S.C.; methodology, N.W. and L.X.; software, N.W. and F.F.; investigation, N.W. and S.C.; data curation, N.W. and F.F.; formal analysis: N.W.; writing—original draft preparation, N.W.; writing—review and editing, S.C. and L.X.; supervision, S.C. and L.X.; funding acquisition, S.C. All authors have read and agreed to the published version of the manuscript.

**Funding:** The authors appreciate the support from the Projects of the National Natural Science Foundation of China (Project No. 71672020; 72072020).

**Institutional Review Board Statement:** Not applicable.

**Informed Consent Statement:** Not applicable.

**Data Availability Statement:** The data presented in this study are available on request from the corresponding author.

**Acknowledgments:** We express our gratitude to the anonymous reviewers and also thank the NSFC for their financial support.

**Conflicts of Interest:** The authors declare no conflict of interest.

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
