# Peer review of "The Sustainability of Superior Performance of Platform Complementor: Evidence from the Effects of Iterative Innovation and Visibility of App in iOS Platform in China"

_sustainability, doi:10.3390/su13074034_

Round 1

Reviewer 1 Report

It is an exciting paper in terms of the topic. The theoretical part has been written very clearly, and hypotheses have been framed logically and convincingly.

First, I can suggest bringing the literature on “demand-side strategies” into the literature review and theoretical arguments related to the iterative innovation and user-oriented perspective. For example, see the papers at the end of my comments.

Second, and more importantly, I am not entirely convinced about H4 and H5. In H4, the visibility through platform recommendation has a synergic effect, but the visibility through app ranking has a substitution effect on iterative innovation. What is the different mechanism between these two types of app visibility that makes the impact opposite? For example, for H5, you say “user satisfaction is reduced when the expectation is greater than the perceived value”; is this argument not relevant for platform recommendation, which raises the users’ expectations? It would be best if you clarified this much better.

I also have some comments on the empirical part as below:

The definition of the dependent variable is not clear. What do you mean by the daily weighted average? I understand each app has one rank per day. So, what is the weight in computing the weighted average? Or do you mean each app can have various ranks in each day in different lists? If so, again, what is the weight? Please clarify.

On a relevant point, what lists are in your data? I see you mention something called the general list; does it mean the overall rank (i.e., all app categories against each other)? You need to mention the name of the lists and their definition clearly.

Is the top 500 list related to top free, or top paid, or top grossing?

How do you count the number of new functions per app for building the Iterative innovation? Is it simply the number of updates, or do you look more closely into the app new features and functions? Please carefully clarify and provide some examples.

What is the filed of an app that you mention in table 1? Is it the app category?

Which categories are included in your sample? Are games included in the sample or not (sometimes you mention excluding games)?

Please clearly mention using OLS regression analysis (as I suppose you have used this) or, if differently, mention the method you use.

What fixed effects are included in your two-way fixed effect model? It is not clear in table 4.

In section 5-3, you mention, “Therefore, we applied cluster processing to the standard error of the regression coefficient to avoid the issue of endogeneity caused by mutual causality effectively”. Using clustered robust standard errors does not link to the reverse (or what you call mutual) causality, or I am missing something here?

Your first robustness test (from line 430) is very vague; please clarify what you apply much better.

Best of luck

Priem, R.L., Wenzel, M. and Koch, J., 2018. Demand-side strategy and business models: Putting value creation for consumers center stage. Long range planning51(1), pp.22-31.

Kalcheva, I., McLemore, P. and Pant, S., 2018. Innovation: The interplay between demand-side shock and supply-side environment. Research Policy47(2), pp.440-461.

Priem, R.L., Li, S. and Carr, J.C., 2012. Insights and new directions from demand-side approaches to technology innovation, entrepreneurship, and strategic management research. Journal of management38(1), pp.346-374.

Reviewer 2 Report

The paper presents an analysis of theory and empirical research on the direct and interactive effects of an app’s iterative innovation and visibility on the sustainability of complementor’s superior performance. The content of the paper fits within the scope of the Journal. It is a new and original contribution. 

  1. The aim of the paper should be clearly stated in one sentence and presented in the introduction.
  2. Please briefly describe in the last paragraph of the INTRODUCTION section the content of each section of the paper and include brief information on methods (one sentence).
  3. Comparisons with other studies have to be provided in the discussion section. Please interpret and describe the significance of your findings in light of what was already known about the research problem being investigated and explain any new understanding or fresh insights about the problem after taking the findings into consideration. Please provide a comparison with other studies.

Reviewer 3 Report

Dear Authors,

Thank you for providing interesting research on iterative innovation and app visibility.

You presented the theoretical background, and I wonder could you discuss the issue of iterative innovations that discourage actual users, but are very attractive to potential users. I am not aware of how large this phenomenon is, however, many users experience this when innovation is getting the app worse. Many applications have failed this way without listening to their users. In search of new users, it lost the old ones, and it did not gain new ones because it lost its visibility. Yik Yak app is an example (not ideal) of how innovative ideas end up with the loss of users (we can discuss if this was an iterative innovation or disruptive innovation). Could you discuss this issue in the theoretical background more deeply?

I think it is important what you wrote in lines 332 to 334. From the iterative innovation point of view, game apps are very interesting. I understand your approach (using the Qimai database), however, I would like to see, at least, some discussion on how different the game apps might be in the context of iterative innovation and visibility. It is because game apps are a large part of the mobile application market.

I do not understand some of your choices. One of them is why you mix variables. As I understood, independent variables are excluding games, and one of your control variables is including games. Could you explain your choice?

Another question is related to the measurement procedure. You have measured iterative innovation as the number of new functions and new content in a monthly app update. Please refer to the removal of certain functions (e.g. functions not used or useless). Is removing a feature that was found not to be preferred by users count as iterative innovation?

Table 2. It is not easy for a reader to remember what are the units in which you measure these variables. You have to go back to the previous table to see what unit was meant. It would be easier if in Table 2 there were also units in which, for example, the average was calculated. Also, it is not clear what is “measure” in Table 2. Is it an average for all apps? I assume this is average because there are SD and extreme values. However, could you be clear on this?

You have said that apps were categorized in 22 fields. Why you use such categorization if you don’t refer to it in the results? It would be great for the potential reader to find in your research how different are education apps and medical apps in terms of sustainability of superior performance.
